# Fine-Grained Permeable Surface Mapping through Parallel U-Net

**DOI:** 10.3390/s24072134

**Published:** 2024-03-27

**Authors:** Nathaniel Ogilvie, Xiaohan Zhang, Cale Kochenour, Safwan Wshah

**Affiliations:** 1Vermont Artificial Intelligence Laboratory (VaiL), Department of Computer Science, University of Vermont, Burlington, VT 05404, USA; nathaniel.ogilvie@uvm.edu (N.O.); xiaohan.zhang@uvm.edu (X.Z.); 2Spatial Analysis Laboratory (SAL), University of Vermont, Burlington, VT 05404, USA; cale.kochenour@uvm.edu

**Keywords:** permeable surface mapping, impervious surface mapping, arid environment, aerial imagery, cross-domain adaptation, image segmentation, U-Net

## Abstract

Permeable surface mapping, which mainly is the identification of surface materials that will percolate, is essential for various environmental and civil engineering applications, such as urban planning, stormwater management, and groundwater modeling. Traditionally, this task involves labor-intensive manual classification, but deep learning offers an efficient alternative. Although several studies have tackled aerial image segmentation, the challenges in permeable surface mapping arid environments remain largely unexplored because of the difficulties in distinguishing pixel values of the input data and due to the unbalanced distribution of its classes. To address these issues, this research introduces a novel approach using a parallel U-Net model for the fine-grained semantic segmentation of permeable surfaces. The process involves binary classification to distinguish between entirely and partially permeable surfaces, followed by fine-grained classification into four distinct permeability levels. Results show that this novel method enhances accuracy, particularly when working with small, unbalanced datasets dominated by a single category. Furthermore, the proposed model is capable of generalizing across different geographical domains. Domain adaptation is explored to transfer knowledge from one location to another, addressing the challenges posed by varying environmental characteristics. Experiments demonstrate that the parallel U-Net model outperforms the baseline methods when applied across domains. To support this research and inspire future research, a novel permeable surface dataset is introduced, with pixel-wise fine-grained labeling for five distinct permeable surface classes. In summary, in this work, we offer a novel solution to permeable surface mapping, extend the boundaries of arid environment mapping, introduce a large-scale permeable surface dataset, and explore cross-area applications of the proposed model. The three contributions are enhancing the efficiency and accuracy of permeable surface mapping while progressing in this field.

## 1. Introduction

Permeable surface mapping aims to identify surface materials that percolate water, which is a critical component of various environmental and civil engineering applications, including urban planning [1], stormwater management [2], and groundwater modeling [3]. Traditionally, this task involves manually classifying different levels of permeability pixel-wise, which can be labor-intensive and time-consuming. With the development of deep learning computer vision models, semantic segmentation can be one of the potential solutions to increase the efficiency of permeable surface mapping. There are many different ways to implement semantic segmentation, as can be seen in this review of Deep Learning Techniques applied to semantic segmentation [4].

While many existing papers [5,6,7,8] studied aerial image segmentation, permeable surface mapping is rarely explored due to difficulties such as (1) lack of a comprehensive dataset that is used for training and evaluation, (2) heavily unbalanced classes, and (3) hard-to-distinguish classes, for example, permeable brown dirt and the semi-permeable brown road. Some existing methods addressed the mapping of permeable surfaces by binary semantic segmentation. For example, ref. [9] used the inception architecture to classify permeable and impervious surfaces. An alternative approach of binary classification of permeable and impermeable surfaces involves reclassifying pre-existing maps [10]. However, fine-grained permeable surface mapping is rarely studied.

To tackle the above-mentioned difficulties, in this research, our goal is to predict fine-grained permeable surfaces by applying semantic segmentation models, particularly a parallel U-Net [11] model. Our approach involves a two-step process for analyzing aerial imagery. The first stage subjects the input imagery to permeable and non-completely permeable surfaces by a binary segmentation model. Subsequently, the permeable area of the same imagery is further segmented into four remaining levels of permeability. This division allows the second stage to fine-tune its predictions on the fine-grained permeable areas, which encompasses the various permeability levels. The outputs from both models are then merged to produce the final output image. As evidenced in our extensive experiments, this novel parallel U-Net enhances the accuracy of the mapping results, particularly in scenarios where the available dataset is small and heavily weighted toward a single category.

Moreover, this paper presents a novel permeable surface segmentation dataset. This dataset captures three counties in the US. Thus, it is able to evaluate the cross-area performance of the model. The first portion of the dataset comprises aerial images that cover San Mateo County. The second portion of the dataset is from Alameda and Contra Costa Counties. Pixel-wise fine-grained permeable labeling of five distinct classes was also provided for all of the aerial images collected. To our knowledge, this is the first large-scale comprehensive permeable surface dataset in this field. By introducing this dataset, we expect it to advance research on permeable surfaces and inspire future research in this area.

Another goal of our research is to create a model that is more generalized across different domains. Domain adaptation in the context of deep neural networks is a powerful technique that allows the transfer of knowledge gained from one geographical location, or domain, to another [12,13]. Often, data collected at one location may exhibit differences in terms of terrain, climate, vegetation, and even sensor characteristics compared to another location. As a result, a model trained on data from one area may not perform effectively when applied to an unseen area. To this end, we designed a cross-area experiment to demonstrate the generalization of our proposed model. From the experiments, our parallel U-Net model performs better across domains compared to the baseline U-Net.

Our research contributions can be summarized as three-fold:We tackle the problem of permeable surface mapping by proposing a two-stage U-Net semantic segmentation deep learning model. Our novel model is demonstrated to accurately annotate multi-class permeable surfaces.A novel permeable surface dataset with fine-grained permeable-level labeling is collected in three distinct areas of the United States of America. To the best of our knowledge, it is the first dataset with multi-level permeable surface annotations.To advance the application of our model in a real-world environment, a cross-area evaluation protocol is designed for the novel dataset. We also thoroughly studied and attributed the effect of different training techniques on cross-area performance.

## 2. Related Work

### 2.1. U-Net-Based Image Segmentation

U-Net [11] was first proposed for precise biomedical image segmentation. It features an encoder–decoder architecture with skip connections to preserve spatial information. Expanding on the U-Net architectures, Liu et al. proposed a cascading U-Net that was applied for Brain Tumor Segmentation, where the architecture involved passing the output from the first U-Net to the subsequent one [14]. Cao et al. proposed Swin-UNet, a U-Net [11] like a swin-transformer-based model, for medical imaging segmentation [15]. Although these models share a similar architecture, they are being applied to different segmentation tasks. Zhang et al. studied lane mark enhancement by adopting a dual U-Net architecture to predict the binary mask of the degraded lane mark regions [16]. In our research, our approach is based on a parallel U-Net model to specifically identify permeable surfaces from aerial imagery. Most U-Net architectures focus on improving a single U-Net [17,18,19,20,21,22]. Their research often involves adjustments to the convolution mechanism [18], variations in datasets, or the implementation of distinct sampling and attention methods. For example, Sun et al. improved U-Net by incorporating a residual encoder into the U-Net architecture [17]. Nölke et al. used a U-Net for the binary classification of palm tree locations [19]. Hordiiuk et al. combined Xception [23] and U-Net to evaluate ship locations on the water [20]. Wang et al. proposed to segment forest fire smoke from aerial images with a customized U-Net [22]. Nabiee et al. adopted a hybrid U-Net with multiscale feature fusion and a multi-scale skip connection to map war destruction seen in aerial imagery [21].

In summary, in this work, our problem requires fine-grain evaluation on an unbalanced dataset, and we propose a novel parallel U-Net that can focus solely on the fine-grain permeable surface mapping. To achieve this goal, we collect and label a novel permeable surface dataset. These are addressed by training on a new dataset not available to other models that have two different areas of permeable surface maps and by enhancing aerial segmentation by extracting features from an infrared fourth band. Finally, we implement a parallel U-Net architecture that enables fine-grain segmentation and improved cross-area support.

### 2.2. Satellite Imagery Analysis

A satellite imagery analysis plays a crucial role in various fields ranging from urban planning to disaster management. Over the past decades, researchers have developed numerous techniques and methodologies for extracting valuable information from satellite imagery. German et al. tackles the algal bloom detection from satellite images by calculating the chlorophyll-a concentration from the temporal satellite image sequence [24]. Rashkovetsky et al. combined multiple U-Nets [11] to detect and segment wildfires from multi-band satellite images [6]. Zhang et al. proposed to identify the location of ground images by comparing the learned features from geo-tagged satellite images [25,26]. Li et al. studied poverty mapping from satellite images by proposing a point-to-region dynamic learning framework that can help people find clean water and sanitation services in low-income countries [27]. Li et al. employed YOLOv3 [28] to monitor the agricultural greenhouse effect from satellite images to improve the effectiveness of the modern agricultural industry [29].

In this paper, we focus on using satellite imagery for the fine-grained permeable surface mapping problem. Different from some of the methods mentioned above, for example, refs. [25,27], which only use color channel information, considering the nature of permeable surface mapping, we also take advantage of the infrared band. Our extensive experiments and ablation studies in Section 4 demonstrate the effectiveness of the infrared band in our proposed model.

### 2.3. Permeable Surface Segmentation Dataset

Several existing datasets have been proposed for permeable surface mapping. For example, Oliazadeh et al. used precipitation measurements to aid in the identification and management of urban runoff [30]. Pelletier et al. assessed the thickness of permeable layers above the Earth’s bedrock, although the maps in this study measure the thickness of soil, intact regolith, and sedimentary deposits [31]. Furthermore, Padmanaban et al. utilized the Land Use and Landcover (LULC) classification system, which consists of eight classes that range from barren land to urban areas to wetlands [32]. Lastly, Tootchi et al. aimed to generate a multi-source groundwater and surface water map by amalgamating multiple datasets [33] such as lakes and surface water bodies [34,35,36], land usage [37], and Hydrological Patterns [38,39].

Our application cannot take advantage of the datasets mentioned above as we have a specific classification system used within our research, especially for permeable surfaces in urban environments. Therefore, we collected and labeled our own dataset, which offers a more in-depth analysis of surface types in an urban setting compared to the previously examined datasets.

## 3. Methods

### 3.1. Background

Consider an input image $I_{a}\in R^{C\times H\times W}$, where *H* and *W* are the image height and width, respectively, and *C* is the number of channels in the input image, which is 4 in our case. The permeable surface mapping task aims to find a function f such that it assigns one type of permeable surface to each pixel. This can be formulated as follows:(1)$$f:I_{a}\in R^{C\times H\times W}\to S\in R^{P\times H\times W},$$
where *S* is denoted as the permeable surface segmentation and *P* is the number of categories of permeable surfaces. In this paper, *P* is fixed to 5. In existing methods, to find the mapping function f, a deep neural network $f_{\theta }$ is trained to learn from paired data of $I_{a}$ and *P* to proximate f, where θ is the parameters of the deep learning model $f_{\theta }$. In practice, as we discussed in Section 2.1, many existing methods choose U-Net [11] architecture for $f_{\theta }$ and training with multi-class cross-entropy loss as follows:(2)$$L_{CE}\left(S,\hat{S}\right)=-\frac{1}{N}\sum_{n=1}^{N}\sum_{p=1}^{P}log\frac{e^{S_{n},p}}{\sum_{i=1}^{P}e^{S_{n},i}}{\hat{S}}_{n,p},$$
where N=H×W is the spatial dimension of the image, *S* is the prediction from $f_{\theta }$, and $\hat{S}$ is the ground truth permeable surface segmentation. However, in our case, there is a heavily unbalanced data distribution, which is the nature of the permeable surface mapping task (as mentioned in Section 4.1). The dominant category can easily overwhelm the training procedure, which causes U-Net to suffer from performance degradation.

### 3.2. Parallel U-Net

To tackle the above-mentioned challenges in permeable surface mapping, we proposed parallel U-Net, which is an enhanced version of U-Net for heavily unbalanced datasets. The key idea of our parallel U-Net is borrowed from the divide-and-conquer concept, which divides the permeable surface mapping into two sub-problems. The first problem involves the intention to segment the fully permeable surface from the other area. The second problem is to perform fine-grained segmentation on semi-permeable surfaces. The final output is simply merging these two predictions. The overview of the proposed parallel U-Net is shown in Figure 1. Formally, this concept can be written as follows:(3)$$f_{\delta }:I_{a}\in R^{C\times H\times W}\to S_{\delta }\in R^{P_{\delta }\times H\times W},$$
(4)$$f_{\gamma }:I_{a}\in R^{C\times H\times W}\to S_{\gamma }\in R^{P_{\gamma }\times H\times W},$$
where $f_{\delta }$ parameterized by δ aims to focus on permeable surface and non-permeable surface segmentation, which takes $I_{a}$ as input and predicts a binary segmentation map while $f_{\gamma }$ with learnable parameter γ predicts the fine-grained permeable surface segmentation for other areas. At the inference stage, the final output *S* can be easily obtained by merging $S_{\delta }$ and $S_{\gamma }$ by
(5)$$S_{i,j}=\left\{\begin{array}{cc} S_{\delta_{i,j}} & \;\mathrm{if}\;S_{\delta_{i,j}}=1\\ S_{\gamma_{i,j}} & \;\mathrm{Otherwise} \end{array}\right.$$
where *i* and *j* are the pixel x-axis and y-axis coordinates. In this simple manner, we can easily merge the output from two models into one single output without any extra calculation.

### 3.3. Training Objectives

During the training phase, $f_{\delta }$ and $f_{\gamma }$ are trained separately for simplicity. Specifically, due to $f_{\delta }$ being only responsible for binary class segmentation, we only adopt binary cross-entropy loss,
(6)$$L_{BCE}={\hat{S}}_{\delta }log\left(S_{\delta }\right)+\left(1-{\hat{S}}_{\delta }\right)log\left(1-S_{\delta }\right).$$

For training $f_{\gamma }$, we adopt the multi-class cross-entropy loss in Equation (2) but ignore the fully permeable surface class as it is handled by $f_{\delta }$.

## 4. Experiment

### 4.1. Dataset

Our dataset contains two main portions. The first portion is collected from San Mateo County, and it is used for training and testing on same-area experiments. The aerial image data contain four channels. The second portion is collected from Alameda and Contra Costa Counties. The aerial imagery contains four channels; the first three channels represent the normal RGB data and the last channel corresponds to the infrared spectrum. For all the aerial images in our dataset, we annotate the permeable surfaces into five different levels, namely, buildings, Other Paved Surfaces, Other Dirt/Gravel Surfaces, paved roads, and all other fully permeable surfaces. The annotation process is in a hybrid style. First, object-based image analysis techniques (OBIA) were used to extract information from land cover using color, tone, texture, pattern, location, size, and shape followed by a classification process to estimate the category of permeable surfaces. Then, a detailed manual review of the dataset was carried out and all observable errors were corrected. Due to the limited capacity of the model and training resources, we tiled the raw large aerial images and the label data into smaller sizes of 512×512. In this manner, we ended up with 2178 images in San Mateo County; specifically, 1465 of them are for training and 713 of them are for testing purposes. Similarly, we ended up with 747 images in Alameda and Contra Costa, which are all for cross-area evaluation. Some randomly sampled processed aerial images as well as their corresponding labels are presented in Figure 2. A summary of the pixel distribution for each class is shown in Table 1 for both San Mateo and Alameda and Contra Costa Counties. This table clearly demonstrates that the data contain a large amount of fully permeable surfaces but only a minimal amount of dirt/gravel surfaces, which is heavily unbalanced in both San Mateo and Alameda and Contra Costa regions.

### 4.2. Implementation Details and Experiment Setup

We conducted our experiments on the dataset we collected as discussed in Section 4.1. Recall that our dataset has two portions, the San Mateo County and Alameda and Contra Costa Counties. The training set and same-area testing set are from San Mateo County and cross-area testing is from Alameda and Contra Costa Counties. For evaluation, we choose two popular metrics, the F1 score and Intersection-Over-Union (IOU) values. Our model is implemented in Python with the PyTorch 1.9.0 (https://pytorch.org/ (accessed on 5 June 2023)) library and TorchSat 0.0.1 (https://github.com/sshuair/torchsat (accessed on 16 June 2023)) library. We adopted ResNet-152 [40] as the default backbone for our proposed parallel U-Net. Common data augmentations such as random flips and random color jittering are applied during training. The proposed parallel U-Net was trained with the ADAM optimizer [41] with a learning rate of 0.001 and a batch sizer of 32. We trained the model for 100 epochs with a cosine learning rate scheduler. The model was trained on a single Nvidia V100 GPU. The following experiments in this section are performed with the above-mentioned hyper-parameters unless we specify otherwise.

### 4.3. Main Results

To demonstrate the effectiveness of our proposed parallel U-Net, we benchmark it with two baseline methods, the original U-Net [11], and segmenter [42], which is a transformer-based segmentation module. To enable it, the segmenter was used on our collected dataset with infrared data. We added an extra branch to extract the infrared information before the scale product operation in the segmenter. The result is presented in Table 2. As demonstrated in this table, the segmenter is collapsing due to the heavily unbalanced data distribution. Thus, we did not show the cross-area results of the segmenter model. Another baseline U-Net achieves a reasonable result on both same-area and cross-area experiments. However, we can see that our proposed parallel U-Net exceeds U-Net on both the same-area and cross-area experiments. Specifically, our parallel U-Net significantly outperforms the original U-Net on cross-area benchmarks.

### 4.4. Ablation Study

To study the effectiveness of our parallel U-Net model, we conducted several ablation studies to demonstrate its capacity, such as ablation on data augmentation, infrared input, and backbone feature extractors.

#### 4.4.1. Ablation Study on Data Augmentation

Data augmentation plays an important role in semantic augmentation. Similarly, as discussed in Section 4.2, we adopted several data augmentations in training our parallel U-Net. In this section, we conducted experiments to analyze the efficacy of data augmentation methods in permeable surface mapping. First, we compare the results of training with or without data augmentation on both same-area and cross-area benchmarks as shown in Table 3. As indicated in this table, data augmentation improves both same-area and cross-area performance on both the baseline U-Net model and our parallel U-Net model. Noticeably, in the cross-area experiment of our proposed parallel U-Net model, there is a significant improvement from without augmentation to with augmentation. This might be caused by the parallel design that our fine-grained segmentation module $f_{\gamma }$ can learn more rich features for augmented semi-permeable surfaces.

To further study the attributes of each type of data augmentation in the training phase, we conducted another experiment to analyze the contribution of two main data augmentation methods, flipping and color jittering (i.e., randomly adjusting the contrast, brightness, and hue), to the proposed parallel U-Net model. The results are presented in Table 4. As we can see from this table, without any augmentation, the model shows a poor performance. After adding horizontal flipping and verticle flipping to the model, the same-area performance decreases a little bit but the cross-area increases by a substantial amount. Moreover, we added color jittering to the data augmentation. We found that color jittering helps both same-area and cross-area benchmarks to improve the F1 score and the IOU score. In summary, data augmentation is important for the permeable surface mapping segmentation task. From the experiment, it is recommended to include both flipping and color jittering to achieve the best results.

#### 4.4.2. Ablation Study on Infrared Band

Different from other satellite image tasks, infrared information is essential in permeable surface mapping. Thus, in this section, we study the importance of infrared information in the proposed parallel U-Net model as shown in Table 5. We experimented on the baseline U-Net and our proposed parallel U-Net. As indicated in the results, infrared band information is critical for both same-area and cross-area benchmarks. Especially in cross-area benchmarks, we can see it improve both baseline U-Net and our proposed parallel U-Net by a large margin. This illustrates that the infrared band contains more invariant information in unseen scenarios.

#### 4.4.3. Ablation Study on Backbone Networks

Finally, we studied the effect of the backbone network on the performance of permeable surface mapping tasks. We varied the backbone feature extractor for the proposed parallel U-Net. We choose four different backbone architectures, namely, ResNet-34, ResNet-50, ResNet-101, and ResNet-152 (default) [40]. The results are presented in Table 6. As we expected, with more convolution layers and the deeper the network, the more accuracy on both same-area and cross-area benchmarks. However, we can also observe that there is a marginal improvement from ResNet-101 to ResNet-152, which suggests that the model almost converges while using ResNet-101 as the backbone. Thus, it is worth considering using ResNet-101 or even using ResNet-50 as the backbone network to balance the accuracy and size of the model in real-world deployment.

### 4.5. Qualitative Analysis

In conducting a qualitative analysis of the data generated through our implementation of the parallel U-Net, there is evidence that has been observed that parallel U-Net can accurately predict the fine-grained permeable surface mapping under different conditions as shown in Figure 3. The parallel architecture has demonstrated enhanced capabilities in isolating and categorizing permeable features amid semi- or non-permeable surroundings.

One issue arises in areas where trees overlap roads, presenting challenges in accurately delineating and distinguishing these features. This is highlighted in Figure 4 in the red box. The intricate nature of such scenarios demands human intervention to ensure precise mapping, as the U-Net output struggles to discern the overlapping elements in the input image. While the parallel U-Net has improved the accuracy of the output when compared to the baseline U-Net as in Section 4.3, the need for manual intervention for the tree–road overlapping issue highlights the importance for future research in this direction and pushes the real-world deployment of fine-grained permeable surface mapping models.

### 4.6. Discussion

The findings presented in Section 4 demonstrate an enhancement in the performance of our parallel U-Net model compared to the baseline U-Net and transformer-based models, particularly in the cross-area evaluation in Alameda and Contra Costa Counties. This improvement demonstrates an outstanding generalization of our proposed parallel U-Net across diverse geographical regions. Specifically, the mean F1 scores and IoU metrics consistently show increased performance for the parallel U-Net in both augmented and non-augmented scenarios, indicating its effectiveness in accurately classifying permeable features in areas with varying characteristics.

In the specific context of Alameda and Contra Costa Counties, the parallel U-Net excels in capturing fine-grained details within the aerial images. This is shown in the higher mean F1 scores and IoU values compared to the baseline U-Net. The improved performance suggests that the parallel architecture can more effectively discern and categorize fine-grained features, providing a detailed and nuanced segmentation of permeable surfaces. This capability is beneficial for applications that demand a higher level of precision, such as urban planning or environmental monitoring in new regions. It is important to acknowledge that while the parallel U-Net creates improvements compared to the baseline U-Net, it may still require manual fixup in certain scenarios such as the tree–road overlapping example shown in Section 4.5. The overall performance boost, especially in cross-area evaluations and fine-grain pixel determination, means that the parallel U-Net is a promising solution for permeable surface applications.

One limitation of the proposed parallel U-Net is the computational efficiency, since the two sub-modules, $f_{\delta }$ and $f_{\gamma }$, are identical. However, $f_{\delta }$ is for a relatively simple task (identifying permeable vs. non-permeable surfaces). Thus, in real-world deployment, it is feasible to prune the weight in $f_{\delta }$ to speed up the inference speed.

## 5. Conclusions and Future Works

In conclusion, permeable surface mapping stands as an essential component in environmental and civil engineering applications, with implications for urban planning, stormwater management, and groundwater modeling. Permeable surface mapping has traditionally been a labor-intensive manual classification process. The use of deep learning and semantic segmentation offers a new solution to enhance the efficiency of this mapping task. While previous studies have explored various semantic segmentation techniques, the field of permeable surface mapping in arid environments has remained unexplored. The first complication was the lack of data for training and evaluation; then, within the dataset, there are large class imbalances, and the difficulty of distinguishing between similar surface categories, particularly in arid regions where the color palettes of permeable and non-permeable surfaces often overlap.

In this paper, we introduce a parallel U-Net for fine-grained permeable surface mapping. Our model leverages the successful U-Net model but innovates a novel parallel structure to tackle the heavily unbalanced data distribution in permeable surfaces. This is achieved through the first U-Net deciding between fully permeable surfaces and non-permeable surfaces. Then, the second U-Net categorizes the non-permeable surfaces into various levels of fine-grained permeability. Merging the outputs of these two steps results in higher accuracy and more generalizable maps, particularly in cases where the dataset is limited and dominated by a single category. For future works, considering the different types of terrain and the change in appearance of the land surface at different times of the year, it is feasible to include such data in our proposed dataset. Thus, it can further facilitate future research on permeable surface mapping and attract more researchers to dedicate to this field.

## Figures and Tables

**Figure 1 sensors-24-02134-f001:**
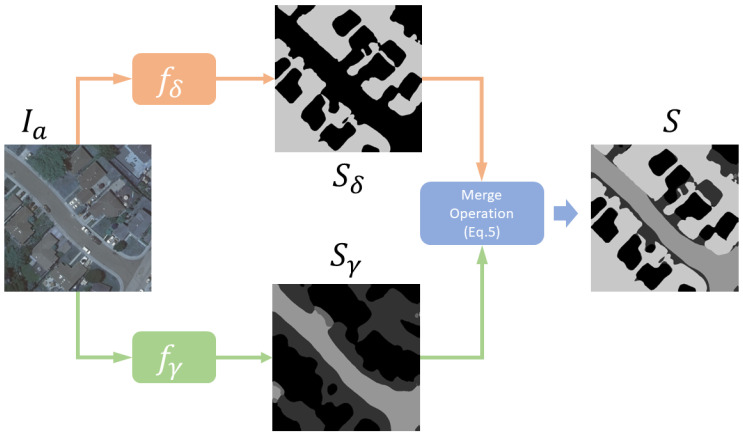
The proposed parallel U-Net architecture. The orange components denote the binary permeable vs. non-permeable segmentation module $f_{\delta }$ and the green components denote the fine-grained permeable surface segmentation module $f_{\gamma }$. The final prediction *S* is merged from these two outputs.

**Figure 2 sensors-24-02134-f002:**
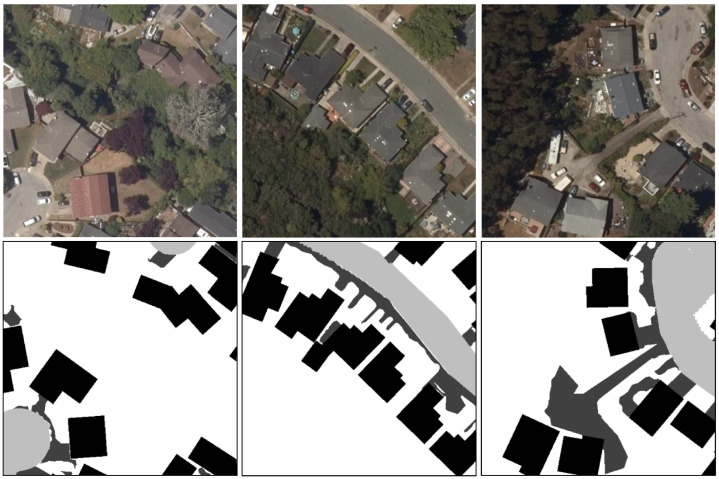
Three randomly sampled data pairs from our permeable surface mapping dataset in the San Mateo region. The images in the first row are aerial imagery in the RGB band. The second row shows the corresponding permeable surface mapping labeling. White means the most permeable surfaces. Black stands for least permeable surfaces.

**Figure 3 sensors-24-02134-f003:**
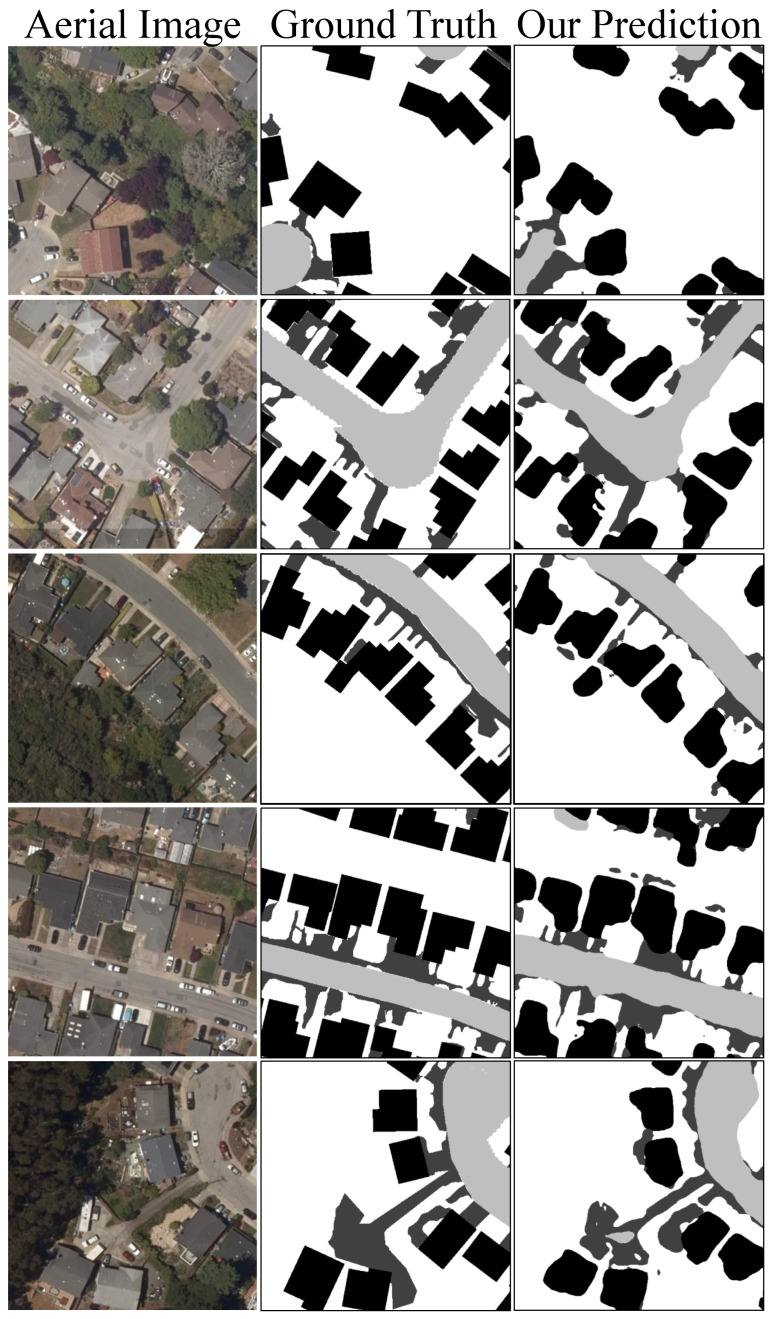
Five random samples’ prediction from our proposed parallel U-Net. From left to right is the input aerial image in the RGB band, the ground truth permeable surface mapping, and our predicted permeable surface mapping.

**Figure 4 sensors-24-02134-f004:**
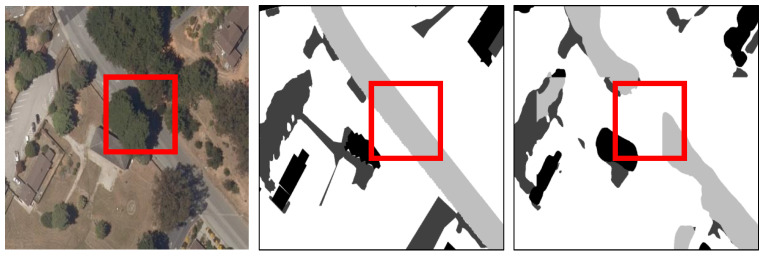
Challenging scenarios when trees and paved road overlap. From left to right is the aerial image, ground truth permeable surface, and our prediction. As shown in the red bounding boxes, this overlapping makes our model fail in predicting a continuous paved road; instead, it predicts fully permeable surfaces since the trees are on top of the road in the aerial image.

**Table 1 sensors-24-02134-t001:** Statistics of each category of the collected dataset.

Region	Buildings	Other Paved Surfaces	Other Dirt/Gravel Surfaces	Paved Roads	Other Fully Permeable Surfaces
San Mateo	11.36%	8.79%	0.29%	9.24%	70.31%
Alameda and Contra Costa	4.06%	5.48%	1.125%	2.47%	86.86%

**Table 2 sensors-24-02134-t002:** Main Results Comparing our Parallel U-Net to Baseline U-Net and Segmenter. The best results are shown in **bold** font.

	Same-Area		Cross-Area	
Method	Mean F1 Score	IOU	Mean F1 Score	IOU
Segmenter	0.1597	0.1329	-	-
U-Net	0.6573	0.5670	0.3723	0.2491
Ours	**0.6800**	**0.5799**	**0.4266**	**0.2769**

**Table 3 sensors-24-02134-t003:** Ablation results of data augmentation on U-Net and the proposed parallel U-Net. The best results are shown in **bold** font. “AUG” stands for data augmentation.

	Same-Area		Cross-Area	
Method	Mean F1 Score	IOU	Mean F1 Score	IOU
U-Net w/o AUG	0.6611	0.5746	0.4080	0.2698
Ours w/o AUG	0.6622	0.5719	0.4107	0.2763
U-Net	0.6573	0.5670	0.3723	0.2491
Ours	**0.6800**	**0.5799**	**0.4266**	**0.2769**

**Table 4 sensors-24-02134-t004:** Ablation results of flipping and color jittering data augmentation in training.

	Same-Area		Cross-Area	
Augmentation	Mean F1 Score	IOU	Mean F1 Score	IOU
None	0.6658	0.5775	0.3827	0.2324
Flip	0.6622	0.5719	0.4107	0.2763
Flip and Color Jitter	0.6800	0.5799	0.4266	0.2769

**Table 5 sensors-24-02134-t005:** Ablation study on the infrared band in baseline U-Net and our proposed parallel U-Net.

	Same-Area		Cross-Area	
Method	Mean F1 Score	IOU	Mean F1 Score	IOU
U-Net w/o Infrared	0.6457	0.5526	0.3566	0.2097
U-Net w/ Infrared	0.6573	0.5670	0.3723	0.2491
Ours w/o Infrared	0.6562	0.5670	0.3844	0.2497
Ours w/ Infrared	0.6800	0.5799	0.4266	0.2769

**Table 6 sensors-24-02134-t006:** Ablation study of backbone network on our Parallel U-Net.

	Same-Area		Cross-Area	
Backbone	Mean F1 Score	IOU	Mean F1 Score	IOU
ResNet-34	0.6567	0.5679	0.3842	0.2581
ResNet-50	0.6459	0.5542	0.3874	0.2292
ResNet-101	0.6581	0.5635	0.4158	0.2746
ResNet-152	0.6800	0.5799	0.4266	0.2769

## Data Availability

Data are contained within the article.

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
