# Peer review of "Fine-Grained Permeable Surface Mapping through Parallel U-Net"

_sensors, 2024, doi:10.3390/s24072134_

Round 1

Reviewer 1 Report

Comments and Suggestions for Authors

Introducing a groundbreaking methodology, this paper leverages a parallel U-Net model for the meticulous semantic segmentation of permeable surfaces. The process initiates with binary classification, distinguishing between entirely and partially permeable surfaces, followed by a fine-grained classification into four distinct permeability levels. The results underscore the substantial enhancement in accuracy brought about by this innovative approach, particularly evident when dealing with small, imbalanced datasets dominated by a single category. Moreover, the proposed model demonstrates its proficiency in generalizing across a spectrum of geographical domains. The exploration of domain adaptation seeks to transfer knowledge seamlessly from one location to another, effectively addressing challenges posed by varying environmental characteristics. Experimental findings substantiate that the parallel U-Net model consistently outperforms baseline methods when applied across diverse domains. In terms of a comprehensive critique, the paper exhibits the following areas of improvement:

(1) Section 2's referencing style necessitates clarification, with the suggestion to refrain from citing literature using verbs and instead emphasize author information.

(2) The structure of Section 2 requires enhancement, currently featuring only two subsections and multiple instances of referencing numerous literature, as exemplified by "multiple datasets [27–40]."

(3) Sections 3 and 6 could benefit from integration with other chapters, given their relatively limited current content.

(4) Despite the authors' qualitative analysis of the data generated through the parallel U-Net implementation, there is a need to underscore the significance of numerical analysis.

(5) Acknowledgments should not be presented as a standalone section, and the conclusion section should encompass a comprehensive discussion of future research trends.

Comments on the Quality of English Language

Minor editing of English language required.

Author Response

Thank you very much for taking the time to review this manuscript. Please find the detailed responses below and the corresponding revisions/corrections highlighted in track changes in the re-submitted files.

(1) Section 2's referencing style necessitates clarification, with the suggestion to refrain from citing literature using verbs and instead emphasize author information.

Thanks for pointing this out. We modify section 2 by using verbs and highlighting the authors' information. 

(2) The structure of Section 2 requires enhancement, currently featuring only two subsections and multiple instances of referencing numerous literature, as exemplified by "multiple datasets [27–40]."

Thanks for your suggestion. 1. We added section 2.2 Satellite Imagery Analysis to include more related work. 2.  We updated the sentence with multiple references as follows,

Lastly, Tootchi et al. aimed to generate a multi-source groundwater and surface water map by amalgamating multiple datasets [33] such as lakes and surface water bodies [34 36 ], land usage [ 37 ], and Hydrological Patterns [38,39].

(3) Sections 3 and 6 could benefit from integration with other chapters, given their relatively limited current content.

Thanks for your suggestion. We merged these two sections into the Experiment section (section 4) to improve the readability. Specifically, Section 4.1 is the dataset section, and Section 4.6 is the Discussion section.

(4) Despite the authors' qualitative analysis of the data generated through the parallel U-Net implementation, there is a need to underscore the significance of numerical analysis.

We agree that numerical analysis is critical to show the significance of our work. Thus we present the comparison between our parallel U-Net and baseline methods in Table 2. We also conducted extensive ablation studies to illustrate the effectiveness of each component of the proposed model such as data augmentation (Table 3), Infrared band (Table 4), and backbone networks (Table 5).

(5) Acknowledgments should not be presented as a standalone section, and the conclusion section should encompass a comprehensive discussion of future research trends.

Thanks for pointing this out. We updated the Acknowledgment section as a standalone section at the end of the paper. Also, we modified the "Conclusion" section into "Conclusion and Future Works" section to discuss potential future works.

Reviewer 2 Report

Comments and Suggestions for Authors

The article fully reveals the solution to the problem of segmenting satellite images to assess permeability. The authors presented their idea with two parallel segmentation stages, presented experimental data, quantitative and qualitative results. I highly appreciate this work, but I would like to draw attention to the following point, which should be disclosed in more detail in the article:

1. The study reveals the importance of segmenting satellite images based on their water permeability for solving environmental issues.  2. The originality of this article lies in the specific problem it addresses, as existing works only consider the segmentation of aerial photographs and do not address the mapping of a permeable surface. This area presents challenges such as insufficient data sets, imbalanced data, and difficulties with dividing a snapshot into classes. The authors propose a novel approach to solve this problem using the U-Net model.  3. Firstly, the study extends the subject area by introducing an automated solution to segment images based on permeability. Second, it extends the application of the U-Net model to a new problem.  4. The methodology is presented in detail, with the approach and necessary formulas being clearly described. As a side note, I would like to draw attention to the following point:  In the Discussion section, I would like to see more information about the limitations of the presented approach. It would also be interesting to present and process a satellite image with a completely different type of terrain, time of year, since now the images presented have great similarities in type of land and vegetation. It is interesting to present to the reader the possibilities of an approach in completely different conditions. If such a result is not satisfactory, it can be noted that this will become a direction for further research.
5.Yes, the results of the experiments confirm the authors' achievement of the stated goal and the successful resolution of the problem. It is natural that the proposed approach could be further developed and refined through additional research in this field.  6. Yes, all relevant links are available. 7. I would like to request the authors, if possible, to extend the experimental section. It would also be interesting to present and process a satellite image with a completely different type of terrain, time of year, since now the images presented have great similarities in type of land and vegetation. It is interesting to present to the reader the possibilities of an approach in completely different conditions. If such a result is not satisfactory, it can be noted that this will become a direction for further research.   The final conclusion of this article is: The article fully reveals the solution to the problem of segmenting satellite images to assess permeability. The authors presented their idea with two parallel segmentation stages, presented experimental data, quantitative and qualitative results.

Author Response

Thank you very much for taking the time to review this manuscript. Please find the detailed responses below and the corresponding revisions/corrections highlighted in track changes in the re-submitted files.

1. In the Discussion section, I would like to see more information about the limitations of the presented approach

We have updated the discussion section with extra information about the limitations of the model. The relevant updated text is as follows,

One limitation of the proposed parallel U-Net is the computational efficiency. Since the two sub-modules, $f_\delta$ and $f_{\gamma}$ are identical. However, $f_\delta$ is for a relatively simple task (identifying permeable vs. non-permeable surfaces). Thus in real-world deployment, it is feasible to prune the weight in $f_\delta$ to speed up the inference speed.

2. It would also be interesting to present and process a satellite image with a completely different type of terrain, time of year, since now the images presented have great similarities in type of land and vegetation. It is interesting to present to the reader the possibilities of an approach in completely different conditions. If such a result is not satisfactory, it can be noted that this will become a direction for further research.

We concur with the reviewer that experimenting with our model on satellite images with different types of terrain, time of year, and different land covers could be very informative to audiences and further strengthen the papers. However, as we discussed in section 1 and section 2.3, collecting such data requires a great amount of labor, especially for the fine-grained permeable surface labeling. Thus, we add this task as a potential future work in this research field. The relevant text is as follows,

For future works, considering the different types of terrain and the change of appearance of the land surface at different times of the year, it is feasible to include such data in our proposed dataset. Thus, it can further facilitate future research on permeable surface mapping and attract more researchers to dedicate to this field.